# Is weight-adjusted waist index more strongly associated with diabetes than body mass index and waist circumference?: Results from the database large community sample study

Jiabei Wu[1], Jinli Guo[2]*

1 Shanxi Medical University, Taiyuan, China, 2 The Second Hospital of Shanxi Medical University, Taiyuan, China

* guojl_551@163.com

## Abstract

### Background

The uncertainty regarding the correlation between the weight-adjusted waist index (WWI) and diabetes within the National Health and Nutrition Examination Survey (NHANES) necessitates further exploration. As indicators of obesity, the differences in the intensity of association between WWI, body mass index (BMI), and waist circumference (WC) with diabetes are worth exploring. This investigation is undertaken to elucidate the association between WWI and diabetes in the NHANES dataset and to compare the extent to which BMI, WC, and WWI were closely associated with diabetes. Then, choose an obesity index that is more strongly associated with diabetes.

### Methods

A comprehensive cross-sectional stratified survey of 7,973 participants from the 2017–2020 NHANES was conducted. WWI is an anthropometric measure based on WC and weight. The formula is WWI (cm/$\sqrt{\text{kg}}$) = WC/$\sqrt{\text{weight}}$. The association between WWI and diabetes was investigated using weighted multiple logistic regression, smooth curve fitting, stratified analysis, and interaction testing.

### Results

The participants' average age was 50.84±17.34 years, and 50.68% of them were female. The detection rate of diabetes was 15.11%. This positive association was particularly notable among non-diabetic patients. For each unit increase in BMI and WC as continuous variables, the likelihood of developing diabetes in the fully adjusted model increased by 5% (OR = 1.05; 95%CI, 1.03–1.07) and 3% (OR = 1.03; 95%CI, 1.02–1.04), respectively, but for each one-unit increase in WWI, the likelihood of developing diabetes increased by 111% (OR = 2.11; 95% CI, 1.68–2.65). Tests of interactions revealed that in various subgroups, the association between diabetes and WWI remained steady.

**Funding:** The author(s) received no specific funding for this work.

**Competing interests:** The authors have declared that no competing interests exist.

## Conclusions

We analyzed 2017–2020 NHANES data to explore the link between WWI and diabetes, finding a consistent positive correlation. The correlation between WWI and diabetes was stronger than that between WC and BMI. WWI seems to offer better potential aid in disease prevention and diagnosis.

## Introduction

Diabetes is a general term for heterogeneous metabolic disorders [1], the main manifestation of which is chronic hyperglycemia. Diabetes is one of the most common and fastest-growing diseases globally, and is anticipated to impact 693 million adult individuals by 2045, a rise of > 50% from 2017 [2]. The serious complications caused by diabetes are roughly divided into microvascular and macrovascular, the prevalence of the former is much higher than that of the latter [3]. Diabetic foot is the main cause of lower limb amputation [4], which brings great physical and mental pain to patients. In 2015, more than 12% of global health spending was spent on tackling diabetes and its complications, placing a significant burden on the global economy [5]. This shows that diabetes is not only a global health problem, but also a major national economic problem [6]. When diabetes is not controlled, prevented, or delayed, the risks associated with diabetes are significant [7]. It can be seen that early diagnosis and prevention of diabetes is crucial.

The obesity rate for adults in the United States as a whole has reached 39.5% and is increasing [8]. Obesity is considered a facilitator of diabetes and not only increases the risk of developing diabetes but also exacerbates its health risks and complicates its management [9].

BMI (body mass index) redefines "healthy weight" and "unhealthy weight" and is one of the most commonly used diagnostic tools [10, 11]. The majority of research uses BMI to identify obesity, yet BMI is unable to discriminate between muscle and fat mass, nor does it accurately reflect the location of fat [12]. Waist circumference (WC) is one measure of obesity that several studies propose to be used to better identify obesity [13]. WC plays an important role not only in the assessment of metabolic syndrome but also a central role in the detection of insulin resistance [14, 15].

In 2018, Park et al. presented the weight-adjusted waist index (WWI), a new body surface measure for assessing obesity [16]. Computed as the quotient of WC (cm) divided by the square root of weight (kg). An increase in WWI indicates a condition characterized by excessive accumulation of body fat and increased loss of muscle mass, which can directly and concretely assess central obesity [17]. Its stability, reliability, and applicability to other racial and ethnic groups are relevant indicators of "true obesity" that reflect metabolic ill-health [18]. An investigation involving approximately one million Korean adults revealed that, unlike BMI and WC, WWI exhibited a positive correlation with cardiovascular mortality, and was the best predictor of cardiometabolic disease and death risk when WWI and BMI were combined [16]. In a comprehensive cross-sectional analysis, a positive correlation between WWI and the likelihood of increased proteinuria in American adults [8]. Higher WWI values are associated with unhealthy body compartments in community-dwelling adults, such as low muscle mass, low bone mass, and high-fat mass [19]. A 2001–2004 study involving 3,884 people showed that an increase in WWI was associated with an increased risk of erectile dysfunction (ED), and WWI was a better predictor of ED than BMI and WC [20].

Traditional body surface measurements have certain limitations in assessing obesity, and the obesity paradox remains that overweight and obese patients have a better prognosis than non-overweight or non-obese patients in those who develop cardiovascular disease [21]. WWI standardized WC with weight, combining the strength of WC while weakening the link to BMI. Still, these associations have only been validated in East Asian populations [22]. A study conducted in 2023 covering Northeast China found that elevated WWI was significantly associated with the incidence of newly diagnosed diabetes among rural adults in China [23], but the relationship between WWI and diabetes among Americans has not been studied in more depth. Therefore, this study drew on 2017–2020 data from the National Health and Nutrition Examination Survey (NHANES) to delve into the association between WWI and diabetes, comparing the strength of the association between BMI, WC, and WWI and diabetes.

## Materials and methods

### Data sources and sample selection

We executed an exhaustive cross-sectional, stratified survey utilizing data extracted from the biennial NHANES. This dataset encompasses a wide array of information, including potential risk factors and nutrient levels, providing a comprehensive exploration of factors influencing the health of the U.S. population [24]. All subjects participating in the NHANES study strictly followed protocols endorsed by the Ethics Review Board of the National Center for Health Statistics (NCHS ERB) and duly executed informed consent forms [25]. NHANES data from 2017 to 2020 were selected for this study. Among 15,560 participants, 6,328 were under the age of 20, 74 were excluded due to pregnancy, 1,177 lacked WC data, and weight data were missing for 5 participants. Inclusion criteria were met by a total of 7,973 participants for the purpose of this study (Fig 1).

### Exposure variable

WWI is an innovative anthropometric indicator based on WC and weight for the assessment of central obesity. Anthropometric assessments monitor their performance through direct observation, data review, and expert inspector assessment, carried out by skilled health technicians in mobile examination center (MEC) [26]. For subsequent analysis, participants were stratified into quartiles (Q1-Q4) based on their WWI values, according to $10.54 \text{cm}/\sqrt{\text{kg}}$, $11.12 \text{cm}/\sqrt{\text{kg}}$, and $11.70 \text{m}/\sqrt{\text{kg}}$, and were ensured that the number of people in each group was approximately equal. WWI was employed in the analysis as both a continuous and categorical variable, serving as the designated exposure variable for this study. WWI ($\text{cm}/\sqrt{\text{kg}}$) is calculated as follows: $\text{WC}/\sqrt{\text{weight}}$ [27].

### Outcome variable

Excluding instances during pregnancy, have you received a diagnosis of diabetes or hyperglycemia from a medical practitioner or healthcare professional [28]? The participants' affirmative answers led them to be classified as diabetic. Diabetes diagnosed by a physician was an outcome variable in this study.

### Covariates

This study compiled potential confounding covariates that could influence the association between WWI and diabetes. The selected covariates included: age (years), gender, ethnicity (categorized as Mexican American, other Hispanic, Non-Hispanic White, Non-Hispanic Black, or other Race), education level (classified as less than high school, high school, or more

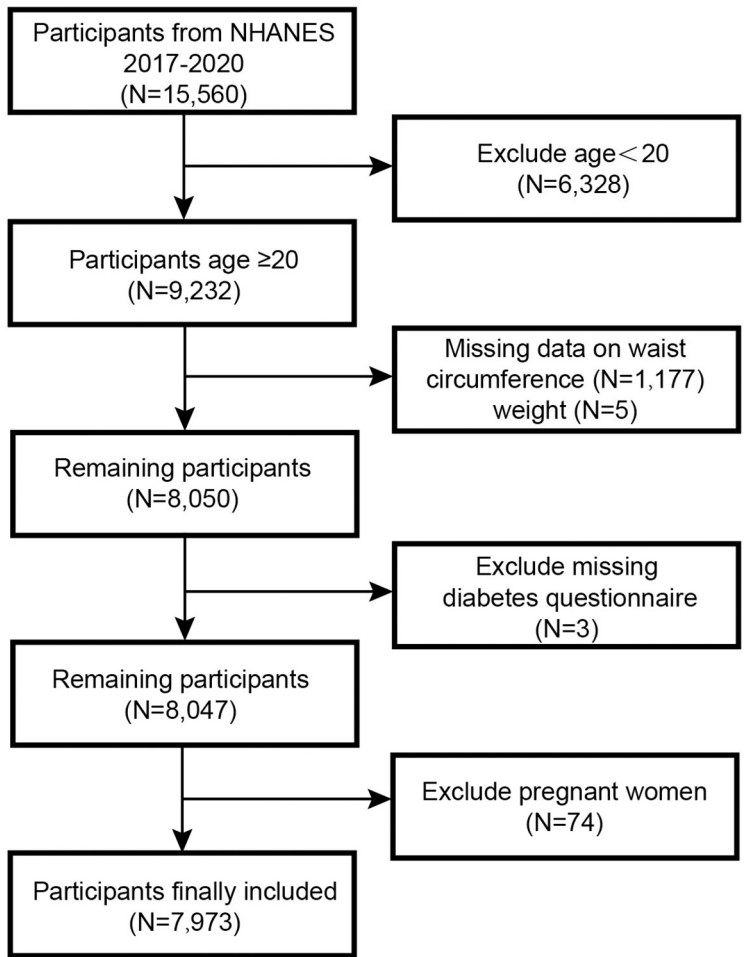

**Fig 1. Flow chart of participants selection.**

than high school), marital status (categorized as married and living with partner, widowed, divorced and separated, never married), smoking status ascertained through the inquiry, 'Have you ever smoked a minimum of 100 cigarettes throughout your lifetime?' (yes/no), hypertension status, moderate activity (yes/no), average alcohol consumption (drinks/day), family income-to-poverty ratio (Family PIR), HDL-C (mg/dL), LDL-C (mg/dL), triglycerides (mg/dL), total cholesterol (mg/dL), BMI ($kg/m^2$), WC (cm), weight (kg), protein intake (gm), total sugars intake (gm), total fat intake (gm), and diabetes status.

## Statistical methods

The continuous variables are delineated through the presentation of mean and standard deviation in the description of the study population, while the categorical variables are reported as frequency (%). We conducted a multivariate logistic regression analysis to explore the correlation between WWI and the presence of diabetes. Model 1 represented an unadjusted model without covariates, while Model 2 incorporated age, gender, and race. The fully adjusted model (Model 3) further controlled for age, gender, race, education level, HDL-C, LDL-C, triglycerides, total cholesterol, Family PIR, total cholesterol, marital status, hypertension status, moderate activity, protein, total sugar, total fat, smoking status, and alcohol consumption.

Models 1 and 2 both included 7,973 participants, with the increase in covariates included in Model 3 resulting in a reduction in numbers to 2,120. We employed weighted multiple logistic regression and smooth curve fitting to investigate the association between WWI and diabetes. Subsequently, stratified analysis and interaction tests were executed.

## Results

### Baseline characteristics of the study population

According to the quartile of WWI, Table 1 displays the clinical and biochemical features of the participants. A total of 7,973 subjects were recruited, with a mean age of 50.84±17.34 years, among which 4,041 cases were female, accounting for 50.68%. The detection rate of diabetes was 15.11%. The overall average of WWI is 11.17±0.86cm/$\sqrt{}$kg. There were statistically significant differences in age, gender, race, education level, marital status, smoking status, alcohol consumption, hypertension, exercise level, BMI, Family PIR, WC, body weight, protein intake, and diabetes according to the WWI quintuple ($p < 0.05$). Non-Hispanic older white women with high blood pressure, higher education than high school, BMI $> 30$kg/m$^2$, light alcohol consumption, larger weight, lower protein content, smoking, less alcohol consumption, no moderate exercise, and no diabetes were more likely to have higher WWI.

### Relationship between WWI and diabetes

To analyze the relationship between WWI and diabetes in depth, we performed multivariate logistic regression analyses. Table 2 demonstrates that, in the fully adjusted model, with each incremental unit increase in WWI, the probability of diabetes occurrence rose by 111% (OR = 2.11;95% CI, 1.68–2.65). The relationship between increased WWI and increased likelihood of diabetes in Model 1 (OR = 2.55; 95% CI, 2.35–2.77, $p < 0.001$), Model 2 (OR = 2.25; 95% CI, 2.04–2.49, $p < 0.001$), and Model 3 (OR = 2.11; 95% CI, 1.68–2.65, $p < 0.001$) were significant. The smooth curve fitting further highlights the significant positive association between WWI and diabetes (Figs 2 and 3). After converting WWI to a categorical variable, the fully adjusted model showed that the highest quartile was almost 5.7 times more likely to develop diabetes than quartile 1 (OR = 5.69; 95% CI, 3.11–10.40). The results in Table 3 show that for each unit increase in BMI and WC as continuous variables, the likelihood of developing diabetes in the fully adjusted model increased by 5% (OR = 1.05; 95%CI, 1.03–1.07) and 3% (OR = 1.03; 95%CI, 1.02–1.04) respectively.

### Subgroup analysis

To further ascertain whether the association between WWI and diabetes was held under different conditions, a subgroup analysis was performed in this study (Fig 4). The interaction test results showed that the association between WWI and diabetes in different subgroups did not show statistically significant differences in each category, and there was a stable existence in age, race, education level, marital status, hypertension, moderate activity status, and smoking status, with no significant effect ($p$ for interaction $> 0.05$).

## Discussion

We collected 2017–2020 NHANES data to explore the potential association between WWI and diabetes. The results of the study revealed that there was a significant positive relationship between WWI and diabetes. For each unit increase in BMI and WC as continuous variables, the likelihood of developing diabetes in the fully adjusted model increased by 5% and 3%, respectively, but for each one-unit increase in WWI, the likelihood of developing diabetes

**Table 1. Basic characteristics of participants by WWI quartile.**

| Characteristics | WWI (cm/√kg) | | | | p-value |
|---|---|---|---|---|---|
| | Q1(<10.54) | Q2[10.54–11.12] | Q3[11.12–11.70] | Q4(≥11.70) | |
| | N = 1,993 | N = 1,993 | N = 1,993 | N = 1,994 | |
| Age(years) | 39.18 ± 14.87 | 48.75 ± 15.48 | 54.83 ± 16.19 | 60.59 ± 15.10 | <0.001 |
| **Gender, n (%)** | | | | | <0.001 |
| Male | 1,176 (59.01) | 1,112 (55.80) | 980 (49.17) | 664 (33.30) | |
| Female | 817 (40.99) | 881 (44.20) | 1,013 (50.83) | 1,330 (66.70) | |
| **Race/ethnicity, n (%)** | | | | | <0.001 |
| Mexican American | 151 (7.58) | 225 (11.29) | 288 (14.45) | 258 (12.94) | |
| Other Hispanic | 166 (8.33) | 211 (10.59) | 207 (10.39) | 226 (11.33) | |
| Non-Hispanic White | 586 (29.40) | 658 (33.02) | 683 (34.27) | 849 (42.58) | |
| Non-Hispanic Black | 695 (34.87) | 514 (25.79) | 495 (24.84) | 414 (20.76) | |
| Other Race | 395 (19.82) | 385 (19.32) | 320 (16.06) | 247 (12.39) | |
| **Education level, n (%)** | | | | | <0.001 |
| Less than high school | 226 (11.34) | 334 (16.78) | 416 (20.89) | 477 (23.97) | |
| High school | 467 (23.43) | 445 (22.36) | 503 (25.26) | 530 (26.63) | |
| More than high school | 1,300 (65.23) | 1,211 (60.85) | 1,072 (53.84) | 983 (49.40) | |
| **Marital status, n (%)** | | | | | <0.001 |
| Married/Living with Partner | 1,062 (53.31) | 1,268 (63.69) | 1,212 (60.90) | 1,091 (54.77) | |
| Widowed/Divorced/Separated | 258 (12.95) | 391 (19.64) | 484 (24.32) | 651 (32.68) | |
| Never married | 672 (33.73) | 332 (16.68) | 294 (14.77) | 250 (12.55) | |
| **Smoked at least 100 cigarettes, n (%)** | | | | | <0.001 |
| Yes | 728 (36.56) | 835 (41.90) | 894 (44.86) | 886 (44.46) | |
| No | 1,263 (63.44) | 1,158 (58.10) | 1,099 (55.14) | 1,107 (55.54) | |
| **Hypertension status, n (%)** | | | | | <0.001 |
| Yes | 361 (18.11) | 664 (33.37) | 899 (45.15) | 1,142 (57.42) | |
| No | 1,632 (81.89) | 1,326 (66.63) | 1,092 (54.85) | 847 (42.58) | |
| **Moderate activity, n (%)** | | | | | <0.001 |
| Yes | 1,045 (52.43) | 863 (43.30) | 773 (38.82) | 610 (30.59) | |
| No | 948 (47.57) | 1,130 (56.70) | 1,218 (61.18) | 1,384 (69.41) | |
| Average alcohol consumption (drinks/day) | 5.26 ± 50.42 | 3.32 ± 26.52 | 3.20 ± 27.66 | 2.91 ± 23.25 | <0.001 |
| Family PIR | 2.76 ± 1.68 | 2.78 ± 1.65 | 2.58 ± 1.60 | 2.34 ± 1.53 | <0.001 |
| HDL-C (mg/dL) | 57.68 ± 16.87 | 53.19 ± 15.77 | 51.95 ± 15.98 | 50.95 ± 14.20 | <0.001 |
| LDL-C (mg/dL) | 105.81 ± 33.41 | 111.81 ± 34.51 | 111.80 ± 38.30 | 105.99 ± 35.91 | <0.001 |
| Triglyceride (mg/dL) | 81.29 ± 58.65 | 114.33 ± 119.23 | 119.90 ± 97.80 | 125.46 ± 90.40 | <0.001 |
| Total cholesterol(mg/dL) | 180.45 ± 37.78 | 189.21 ± 40.13 | 188.82 ± 42.95 | 184.26 ± 41.33 | <0.001 |
| BMI (kg/m$^2$) | 25.33 ± 5.20 | 28.99 ± 6.05 | 31.12 ± 6.86 | 34.25 ± 7.89 | <0.001 |
| Waist circumference (cm) | 85.92 ± 11.24 | 98.00 ± 12.40 | 105.04 ± 13.73 | 114.94 ± 16.12 | <0.001 |
| Weight(kg) | 74.65 ± 17.97 | 82.90 ± 21.01 | 86.27 ± 22.79 | 90.32 ± 25.31 | <0.001 |
| Protein (gm) | 87.76 ± 49.83 | 81.34 ± 41.96 | 77.73 ± 39.39 | 70.03 ± 35.67 | <0.001 |
| Total sugars (gm) | 111.81 ± 84.05 | 108.00 ± 76.24 | 106.01 ± 75.44 | 96.54 ± 66.50 | <0.001 |
| Total fat (gm) | 93.95 ± 55.92 | 87.64 ± 48.31 | 85.96 ± 47.57 | 79.44 ± 45.36 | <0.001 |
| **Diabetes, n (%)** | | | | | <0.001 |
| Yes | 61 (3.06) | 219 (10.99) | 346 (17.36) | 579 (29.04) | |
| No | 1,932 (96.94) | 1,774 (89.01) | 1,647 (82.64) | 1,415 (70.96) | |

Mean ± SD for continuous variables: the P value was analyzed via ANOVA. (%) for categorical variables: the P value was analyzed via the weighted chi-square test.

Abbreviation: Family PIR, the ratio of family income to poverty; BMI, body mass index; HDL-C, High-Density Lipoprotein Cholesterol; LDL-C, Low-Density Lipoprotein Cholesterol; WWI, Weight-adjusted waist index

**Table 2. Association of WWI with diabetes.**

|  | OR (95% CI) | | |
| --- | --- | --- | --- |
| Exposure | Model 1 | Model 2 | Model 3 |
| WWI (m/$\sqrt{}$kg) | 2.55 (2.35, 2.77) | 2.25 (2.04, 2.49) | 2.11 (1.68, 2.65) |
| WWI quartile |  |  |  |
| Quartile 1 | Reference | Reference | Reference |
| Quartile 2 | 3.91 (2.92, 5.23) | 2.95 (2.19, 3.98) | 2.18 (1.20, 3.97) |
| Quartile 3 | 6.65 (5.03, 8.80) | 4.24 (3.16, 5.69) | 3.60 (2.00, 6.47) |
| Quartile 4 | 12.96 (9.87, 17.02) | 8.15 (6.07, 10.94) | 5.69 (3.11, 10.40) |
| P for trend | <0.001 | <0.001 | <0.001 |

In sensitivity analysis, WWI was converted from a continuous variable to a categorical variable (quartile).

Model 1 = No covariates were adjusted.

Model 2 = Age, gender, and race were adjusted.

Model 3 = Age, gender, race/ethnicity, educational level, HDL-C, LDL-C, triglyceride, total cholesterol, Family PIR, total cholesterol, marital status, hypertension status, moderate activity, protein, total sugar, total fat, smoking status and drinking alcohol status were adjusted.

Abbreviation: OR, odds ratio; 95% CI, 95% confidence interval; Family PIR, the ratio of family income to poverty; HDL-C, High-Density Lipoprotein Cholesterol; LDL-C, Low-Density Lipoprotein Cholesterol; WWI, Weight-adjusted waist index.

increased by 111%. In the fully adjusted model, the highest quartile of WWI ($\geq$11.70cm/$\sqrt{}$kg) was associated with 5.69 times the likelihood of developing diabetes compared with Q1 (<10.54cm/$\sqrt{}$kg). Subgroup analysis demonstrates that the association between WWI and diabetes remains consistent across all subgroups. No significant differences are observed among the various subgroups in terms of exposure factors and outcome indicators. Nonetheless, we found in the stable subgroup that education above high school was more likely to influence the relationship between WWI and diabetes than education below high school. Compared with WC and BMI, monitoring WWI can provide better early warning to help identify individuals at risk for diabetes. This helps to take preventive measures, such as diet and lifestyle changes, to reduce the likelihood of diabetes developing. When necessary, the medical staff can develop a personalized treatment plan for the patient and provide effective medical advice.

Several studies have investigated the possible correlation between WWI and diabetes. A study on cardiovascular health was conducted in rural Northeast China followed 9,205 non-diabetic patients for 3 years and analyzed the data using a multivariate logistic regression model to show the adverse effect of increased WWI on diabetes [23]. Muscle-reducing obesity plays a significant part in the metabolism of diabetic patients. A study from South Korea analyzed 515 diabetic patients with SPSS and concluded that WWI was a good anthropometric indicator for predicting diabetic patients [29]. Obesity is the foundation of WWI [30]. According to the accumulation of previous studies, obesity raises the risk of several chronic illnesses, including diabetes [31]. Obesity assumes a unique role in perpetuating the chronic activation of inflammatory pathways, and obesity inflammatory signaling pathways are causally related to insulin resistance [32]. These findings, which are like our own, reflect that WWI can affect the function of multiple systems in the body, indirectly or directly affecting the likelihood of developing diabetes in at-risk individuals. Higher levels of total physical activity have been found to significantly reduce the risk of diabetes through studies of different types of activity at different intensities. A study among U.S. adults noted that diabetes is more prevalent among people with low levels of education [33]. Race, age, and gender differences may exist in the

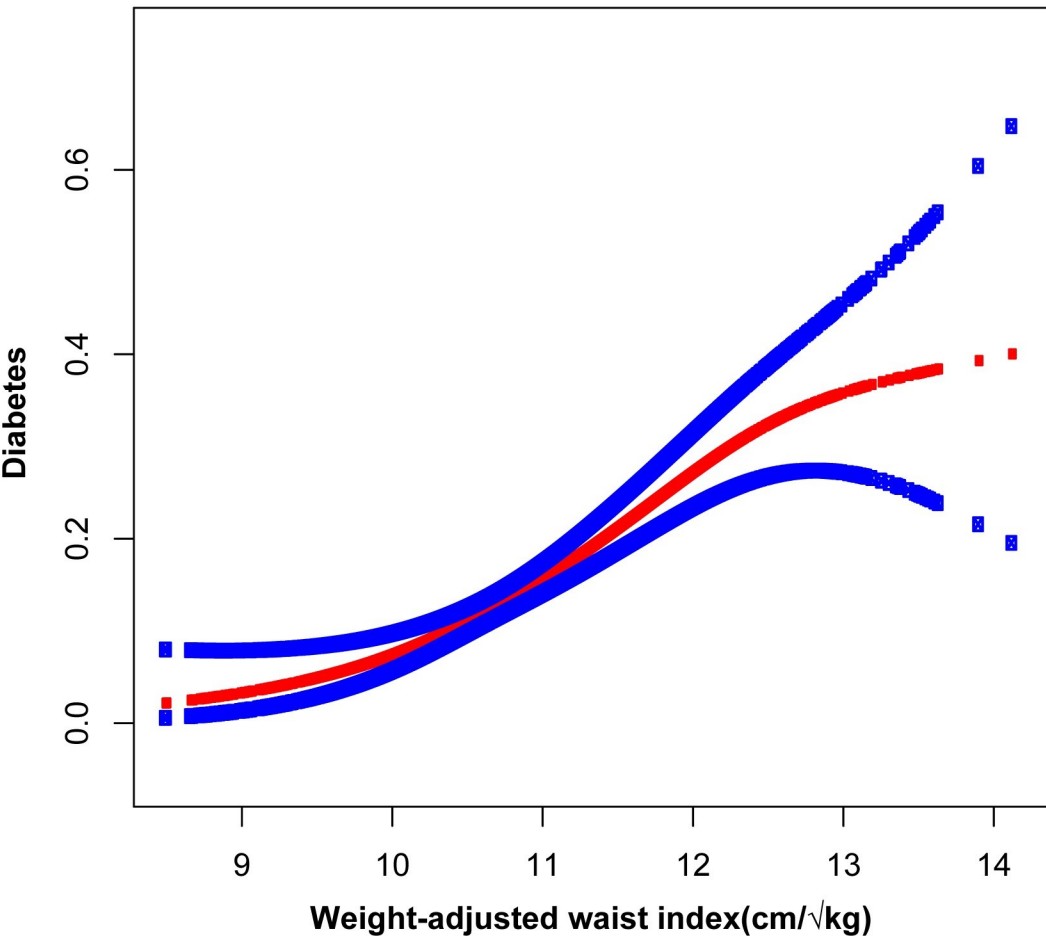

**Fig 2. The association between weight-adjusted waist index and diabetes.** Horizontal coordinates represent WWI, vertical coordinates represent diabetes. The solid red line indicates a smooth curve fit between the variables. Blue bands indicate 95% confidence intervals for the fit. Adjusted variables: Age, gender, race/ethnicity, educational level, HDL-C, LDL-C, triglyceride, total cholesterol, Family PIR, total cholesterol, marital status, hypertension status, moderate activity, protein, total sugar, total fat, smoking status, and drinking alcohol status were adjusted.

predictive power of obesity metrics for diabetes [34–37], but in our stratified analysis, the association between WWI and diabetes was stable across all subgroups.

A study of 240 patients who underwent renal ultrasound noted that hypoechoic perirenal fat was predictive of diabetes [38]. However, this method is not indicated in patients with pregnancy and abnormal renal morphology. The positive correlation between WWI and visceral fat area [39], and the ease of obtaining and calculating the data, provide a unique perspective on the prediction of diabetes, as well as the simplicity of data acquisition and computation, not only provides a unique perspective for the prediction of diabetes, but also combines WC with weight while weakening the association with BMI, and our study may provide preliminary evidence for prospective studies to help identify potential confounders.

Increases in inflammatory markers in the islets are closely linked to obesity. Furthermore, obesity-related inflammation is closely linked to insulin resistance, which in turn causes diabetes [40]. Genomic analysis of blood samples from 26 participants revealed a significant elevation in IL-17 production by CD27−MAIT cells under conditions of diabetes-associated obesity. The dysregulation of the gut microbiome-immune axis in individuals with obesity

## Adjusted mean & 95% CI

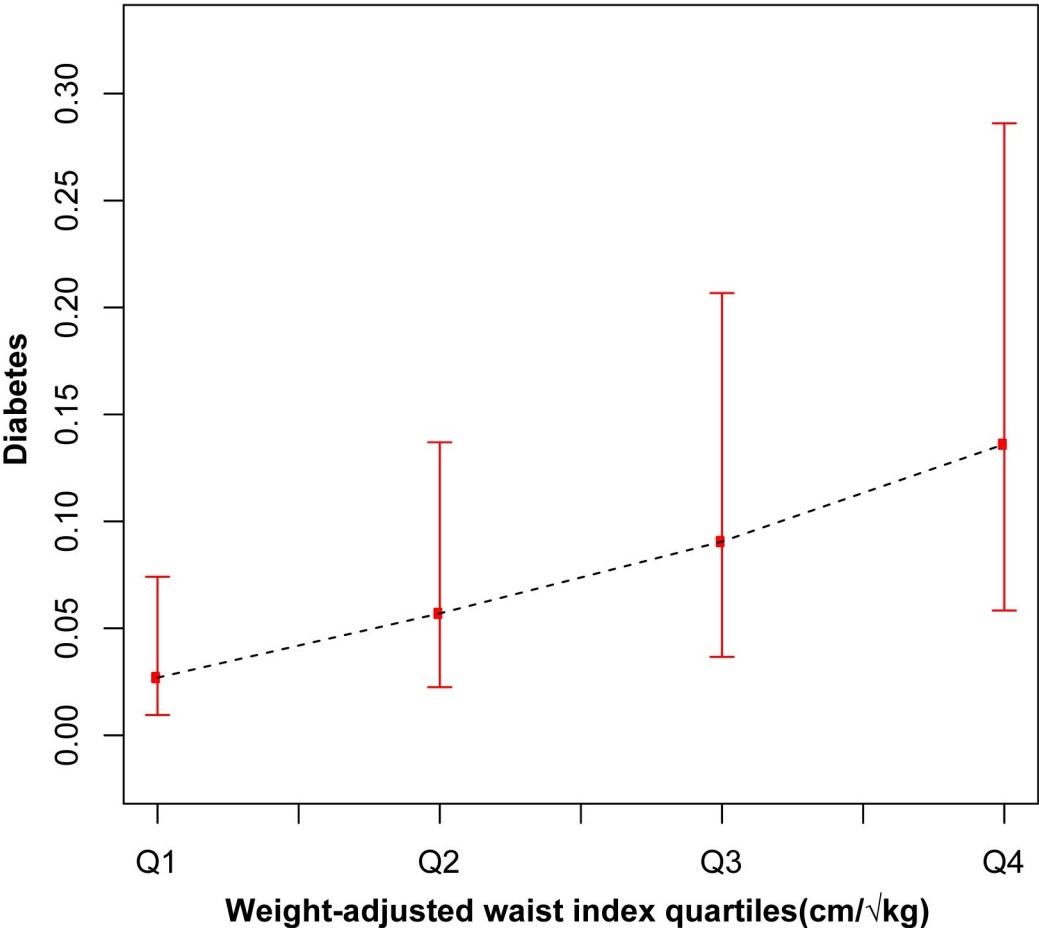

**Fig 3. The association between weight-adjusted waist index quartiles and diabetes.** Adjusted variables: Age, gender, race/ethnicity, educational level, HDL-C, LDL-C, triglyceride, total cholesterol, Family PIR, total cholesterol, marital status, hypertension status, moderate activity, protein, total sugar, total fat, smoking status, and drinking alcohol status were adjusted.

may be a contributing factor to the onset or exacerbation of diabetes [41]. By studying the molecular mechanism of diet-gene interaction, Yang et al. revealed the molecular mechanism of diabetes, pointing out that obesity due to overnutrition is a major risk factor for the development of insulin resistance and diabetes [42]. A single IL-6 or a combination of CRP-IL-6 and CRP-fibrinogen plays an important role in the prediction of diabetes, and the main reason for the increase in inflammatory markers may be obesity, through which diabetes risk increases [43].

There are several significant advantages to our research. We used a large representative sample study, the two most recent data sets were selected for the years, confounding covariables were adjusted to make the results more reliable, and sensitivity analyses were performed. To our knowledge, this study is not only a study evaluating the relationship between WWI and diabetes, but it is also the first study comparing the strength of the relationship between WWI, BMI, and WC with diabetes. Nevertheless, certain limitations exist. First, given the cross-

**Table 3. Association of BMI and WC with diabetes.**

| | OR (95% CI) | | |
|---|---|---|---|
| Exposure | Model 1 | Model 2 | Model 3 |
| BMI (m/√kg) | 1.05 (1.04, 1.06) | 1.07 (1.06, 1.08) | 1.05 (1.03, 1.07) |
| WC (cm) | 1.04 (1.03, 1.04) | 1.04 (1.04, 1.04) | 1.03 (1.02, 1.04) |

Model 1 = No covariates were adjusted.

Model 2 = Age, gender, and race were adjusted.

Model 3 = Age, gender, race/ethnicity, educational level, HDL-C, LDL-C, triglyceride, total cholesterol, Family PIR, total cholesterol, marital status, hypertension status, moderate activity, protein, total sugar, total fat, smoking status and drinking alcohol status were adjusted.

Abbreviation: OR, odds ratio; 95% CI, 95% confidence interval; Family PIR, the ratio of family income to poverty; HDL-C, High-Density Lipoprotein Cholesterol; LDL-C, Low-Density Lipoprotein Cholesterol; BMI, body mass index; WC, waist circumference.

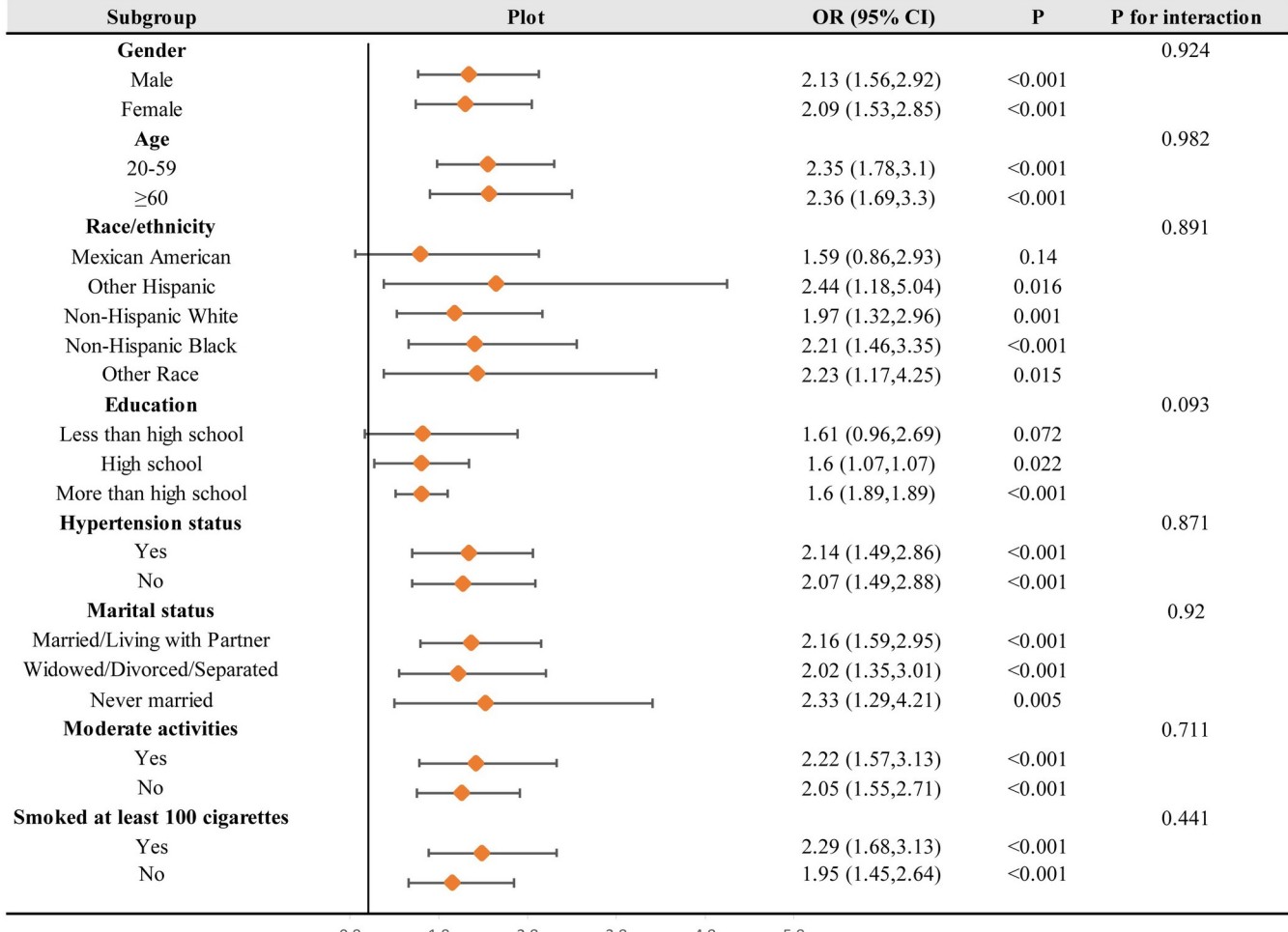

**Fig 4. Subgroups analyses.** In the subgroup analysis stratified by gender, age, race/ethnicity, education levels, smoking status, moderate activity, marital status, and hypertension status, the model is not adjusted for gender, age, race/ethnicity, education levels, smoking status, moderate activity, marital status, and hypertension status, respectively.

sectional design, elucidating the causal relationship between exposure factors and outcomes is challenging. Consequently, the results of this study should be interpreted cautiously, and further clarification of causality necessitates multiple prospective studies. Second, although we included several covariates, we still could not take all covariates about WWI and diabetes into account in the study. Subsequently, we made adjustments for potential confounding variables; however, the presence of residual confounding factors cannot be entirely ruled out. Finally, due to database limitations, our study population was limited to the Americans included in NHANES, and the findings cannot be generalized globally.

## Conclusion

We investigated utilizing the 2017–2020 NHANES data to examine the association between WWI and diabetes. The findings indicate the possibility of a positive correlation between WWI and diabetes, the association between WWI and diabetes was stable in subgroup analyses. Compared with WC and BMI, WWI, as a novel obesity index, is more closely associated with diabetes. This could make an important contribution to the development of preventive detection and care strategies for diabetes.

## Acknowledgments

This research did not receive any specific grant from funding agencies in the public, commercial, or not-for-profit sectors.

## Author Contributions

**Conceptualization:** Jinli Guo.

**Data curation:** Jiabei Wu.

**Formal analysis:** Jiabei Wu.

**Investigation:** Jiabei Wu.

**Methodology:** Jiabei Wu.

**Software:** Jiabei Wu.

**Visualization:** Jiabei Wu.

**Writing – original draft:** Jiabei Wu.

**Writing – review & editing:** Jiabei Wu, Jinli Guo.

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
