## [Decision Letter · Decision Letter 0]

20 Mar 2024

PONE-D-24-02729Is weight-adjusted waist index more strongly associated with diabetes than body mass index and waist circumference?: results from the database large community sample studyPLOS ONE

Dear Dr. Wu,

Thank you for submitting your manuscript to PLOS ONE. After careful consideration, we feel that it has merit but does not fully meet PLOS ONE’s publication criteria as it currently stands. Therefore, we invite you to submit a revised version of the manuscript that addresses the points raised during the review process.

 **This manuscript requires a minor revision****Please thoroughly address all the comments raised by the 2 reviewers**

We look forward to receiving your revised manuscript.

Kind regards,

Fredirick Lazaro mashili, MD, PhD

Academic Editor

PLOS ONE

Journal Requirements:

Additional Editor Comments:

The reviewers have addressed mort of the previously raised concerns. However, there still minor concerns that need to be addressed to improve the article. Please thoroughly address all the comments raised by both the reviewers.

Reviewers' comments:

Reviewer's Responses to Questions

**Comments to the Author**

1. Is the manuscript technically sound, and do the data support the conclusions?

Reviewer #1: Yes

Reviewer #2: Partly

2. Has the statistical analysis been performed appropriately and rigorously? 

Reviewer #1: Yes

Reviewer #2: I Don't Know

3. Have the authors made all data underlying the findings in their manuscript fully available?

Reviewer #1: Yes

Reviewer #2: Yes

4. Is the manuscript presented in an intelligible fashion and written in standard English?

Reviewer #1: Yes

Reviewer #2: Yes

5. Review Comments to the Author

Reviewer #1: In this manuscript, titled "Is weight-adjusted waist index more strongly associated with diabetes

than body mass index and waist circumference?: results from the

database large community sample study" the authors aimed to investigate the strength of association between WWI and diabetes, by comparing it to that of diabetes and BMI and WC. The authors report that WWI has a superior association with DM than that between DM and BMI and/or WC. This is a novel study and the findings add to the ongoing search for better screening surrogates for DM. Despite the novelty, and a variety of positives attributes, minor but very important revisions will improve the quality of the manuscript.

1. While the title explicitly mention the comparators (BMI and WC), the abstract only describes WWI results. It would be good to add a sentence showing the association between DM and WWI was stronger than that between DM and WC and DM and BMI.

2. The main aim of the study comes too late in the introduction. The introduction of DM and its complication is a bit too long delaying the introduction of the main iam. The association/relationship between DM/related factors and WWI, WC and/or WC should appear early on in the introduction.

3. The introduction also did not feature BMI and WC which are explicitly mentioned on the title. It looks like the aim was just to investigate WWI and not comparing its performance/association against WC and BMI.

Reviewer #2: Abstract

Lines 21-23 paraphrase the sentences for readability

Materials and Methods

Exposure variable - Re-write the paragraph. It is not coherent

What was the essence of only recording moderate physical activity?

Results

Table 1 – Amend the intervals used to generate quartiles e.g. value 11.22 is inclusive in Q2 and Q3

Tables 2 and 3. The notes below the tables contain unnecessary definitions of abbreviations that are not used in the tables

Discussion

The is no paragraph that would lead to answering the question featured in the title. Likewise, the discussion section should come up with possible explanations for the observed correlation between WWI and diabetes.

6. PLOS authors have the option to publish the peer review history of their article (what does this mean?). If published, this will include your full peer review and any attached files.

Reviewer #1: **Yes: **Fredirick Mashili

Reviewer #2: **Yes: **Oscar Mbembela

---

## [Author Response · Author response to Decision Letter 0]

27 Mar 2024

Dear Editor and Reviewers, 

We appreciate the opportunity to allow us to revise our manuscript and thanks for the reviewers’ constructive comments and suggestions. We would like to submit our revised manuscript, entitled “Is weight-adjusted waist index more strongly associated with diabetes than body mass index and waist circumference?: results from the database large community sample study” for consideration for publication. In the revised manuscript, we have carefully addressed all comments and questions raised by reviewers point-by-point. We marked the changes in red in the revised paper. We greatly appreciate your time and efforts to improve our manuscript for publication.

Reply to Reviewers 

Reviewer 1 #:

Question 1: While the title explicitly mention the comparators (BMI and WC), the abstract only describes WWI results. It would be good to add a sentence showing the association between DM and WWI was stronger than that between DM and WC and DM and BMI.

Response: Thank the reviewer for pointing out our oversight. The fact that WWI is more relevant than DM and WC has been added on lines 44-46.

Question 2: The main aim of the study comes too late in the introduction. The introduction of DM and its complication is a bit too long delaying the introduction of the main aim. The association/relationship between DM/related factors and WWI, WC and/or WC should appear early in the introduction.

Response: We are very sorry for the delay in the introduction of the main objective due to the cumbersome language. The introduction to DM has now been streamlined and the main purpose introduced in advance, in lines 56-68.

Question 3: The introduction also did not feature BMI and WC which are explicitly mentioned on the title. It looks like the aim was just to investigate WWI and not compare its performance/association against WC and BMI.

Response: Thanks to the reviewer for carefully pointing out our shortcomings, the introduction of BMI and WC has been added in lines 69-76 and marked in red font.

Reviewer 2 #: 

Question 1:Abstract： Lines 21-23 paraphrase the sentences for readability

Response: Sorry, our expression is not clear and accurate enough. We sincerely thank the reviewer for your enthusiastic work. After a double check of grammar and word order, the content has been revised on lines 21-23. 

Question 2: Materials and Methods: Exposure variable - Re-write the paragraph. It is not coherent.

Response: We think this is a good suggestion, and we apologize for our lack of clarity and accuracy. This has been amended in lines 137-146.

Question 3: What was the essence of only recording moderate physical activity?

Response: We greatly appreciate your professional comments on our articles. The questions for moderate activity in NHANES are as follows：In a typical week {do you/does SP} do any moderate-intensity sports, fitness, or recreational activities that cause a small increase in breathing or heart rate such as brisk walking, bicycling, swimming, or volleyball for at least 10 minutes continuously? There are fewer missing parts in this data, and the problems are more detailed and objective. In many similar cross-sectional studies, moderate activity also appears as a covariate[1–3].

Question 4: Results: Table 1–Amend the intervals used to generate quartiles e.g. value 11.22 is inclusive in Q2 and Q3.

Response: Thanks to the reviewer for pointing out our negligence carefully and accurately. We are very sorry and have corrected the relevant content in Table 1 by using open and closed intervals.

Question 5: Tables 2 and 3. The notes below the tables contain unnecessary definitions of abbreviations that are not used in the tables.

Response: Thank you for your careful examination. All abbreviations have been carefully reviewed and revised to ensure that each is necessary in the header or table, and the abbreviations in Tables 2 and 3 and their corresponding content are highlighted in red font.

Question 6: Discussion: The is no paragraph that would lead to answering the question featured in the title. Likewise, the discussion section should come up with possible explanations for the observed correlation between WWI and diabetes.

Response: Thank you very much for your comments on the discussion section. Our study shows that WWI is more strongly associated with diabetes than WC and BMI, and details similar studies on WWI. We have revised these contents in detail in lines 296-311 and 357-360 of the paper.

These comments are all valuable and enable us to greatly improve the quality of our manuscript. Our whole team made meticulous revisions to the manuscript word for word according to the reviewer's opinions, making the manuscript more fit with the title, the content more scientific and orderly, and the text expression clearer and smoother. We tried our best to improve the manuscript. These changes will not influence the content and framework of the paper. We appreciate for Editors/Reviewers’ warm work earnestly and hope that the corrections will meet with approval. Please do not hesitate to contact us with any further questions or recommendations. 

Once again, thank you very much for your comments and suggestions.

Sincerely yours, 

Jinli Guo

References:

1. Liu H, Ma Y, Shi L. Higher weight-adjusted waist index is associated with increased likelihood of kidney stones. Front Endocrinol. 2023;14: 1234440. doi:10.3389/fendo.2023.1234440

2. Cao S, Hu X, Shao Y, Wang Y, Tang Y, Ren S, et al. Relationship between weight-adjusted-waist index and erectile dysfunction in the United State: results from NHANES 2001-2004. Front Endocrinol. 2023;14: 1128076. doi:10.3389/fendo.2023.1128076

3. Wang K, Mao Y, Lu M, Liu X, Sun Y, Li Z, et al. Association between serum Klotho levels and the prevalence of diabetes among adults in the United States. Front Endocrinol. 2022;13: 1005553. doi:10.3389/fendo.2022.1005553

---

## [Decision Letter · Decision Letter 1]

7 May 2024

PONE-D-24-02729R1Is weight-adjusted waist index more strongly associated with diabetes than body mass index and waist circumference?: results from the database large community sample studyPLOS ONE

Dear Dr. Wu,

Thank you for submitting your manuscript to PLOS ONE. After careful consideration, we feel that it has merit but does not fully meet PLOS ONE’s publication criteria as it currently stands. Therefore, we invite you to submit a revised version of the manuscript that addresses the points raised during the review process.

 Please address thoroughly all the comments raised by the third reviewerThis manuscript requires a major revisionPlease ensure that you adhere to all the journal’s requirements 

We look forward to receiving your revised manuscript.

Kind regards,

Fredirick Lazaro mashili, MD, PhD

Academic Editor

PLOS ONE

Additional Editor Comments:

Please address thoroughly all the comments given by the third reviewers. This manuscript requires a major revision. Please adhere to all the journal’s requirements.

Reviewers' comments:

Reviewer's Responses to Questions

**Comments to the Author**

1. If the authors have adequately addressed your comments raised in a previous round of review and you feel that this manuscript is now acceptable for publication, you may indicate that here to bypass the “Comments to the Author” section, enter your conflict of interest statement in the “Confidential to Editor” section, and submit your "Accept" recommendation.

Reviewer #1: All comments have been addressed

Reviewer #3: (No Response)

2. Is the manuscript technically sound, and do the data support the conclusions?

Reviewer #1: Yes

Reviewer #3: Partly

3. Has the statistical analysis been performed appropriately and rigorously? 

Reviewer #1: Yes

Reviewer #3: I Don't Know

4. Have the authors made all data underlying the findings in their manuscript fully available?

Reviewer #1: Yes

Reviewer #3: Yes

5. Is the manuscript presented in an intelligible fashion and written in standard English?

Reviewer #1: Yes

Reviewer #3: No

6. Review Comments to the Author

Reviewer #1: All the previously raised concerns have been sufficiently addressed. The authors have thoroughly reviewed the manuscript and responded to reviewers comments.

Reviewer #3: 1. Introduction

- The introduction is very long and unfocused, the authors should shorten it and clarify the message so it is clear to the reader what is the problem that the authors are addressing.

- The authors make rather outdated statements like "BMI is the new norm for defining .....''

- line 75 and 76 "WC is an essential part of the metabolic syndrome which also involves the detection of insulin resistance''. This sentence is confusing and should be clarified.

- Line 91 " associated with unhealthy body components..." it is body compartments not components

- Line 100 "that the optimal BMI or the oldest elderly may be in the overweight or mildly obese range". This is also a very generalized and unsubstantiated statement.

- Line 112 "The correlation between WWI and diabetes has important clinical significance in promoting patients' physical and mental health, reducing social burden, and rationally allocating medical resources". It is unclear whether the authors are advocating for WWI as a predictor for diabetes or just correlated with diabetes. If it is the latter what is its significance then?

2. Results

- Table 1: Did the authors stratify their study population tot ensure equal numbers among the four quartiles Q1- Q4 (N=1993). This was not indicated in their methods section.

- Table 2: Model 3 N= 2120 while model 1 and 2 have n = 7973 participants. what happened to the missing participants in model 3?

- Figure 2 - 5. There figure legends are just the covariates assessed and not a description of the figures. The figures also lack any key and can not be interpreted.

3. Discussion

- The authors do not really discuss their results in relation to other published literature, the discussion section needs to be re-written. The discussion is also unfocused and delves into topics that are outside the topic of the manuscript (Lines 312 - 353)

7. PLOS authors have the option to publish the peer review history of their article (what does this mean?). If published, this will include your full peer review and any attached files.

Reviewer #1: **Yes: **Fredirick mashili

Reviewer #3: No

---

## [Author Response · Author response to Decision Letter 1]

28 May 2024

Dear Editor and Reviewers, 

We appreciate the opportunity to allow us to revise our manuscript and thanks for the reviewers’ constructive comments and suggestions. We would like to submit our revised manuscript, entitled “Is weight-adjusted waist index more strongly associated with diabetes than body mass index and waist circumference?: results from the database large community sample study” for consideration for publication. In the revised manuscript, we have carefully addressed all comments and questions raised by reviewers point-by-point. We marked the changes in red in the revised paper. We greatly appreciate your time and efforts to improve our manuscript for publication.

Reviewer #3: 1. Introduction

- The introduction is very long and unfocused, the authors should shorten it and clarify the message so it is clear to the reader what is the problem that the authors are addressing.

Response: We greatly appreciate your professional comments on our articles. Based on your comments, we have streamlined the introductory section by eliminating repetitive and irrelevant remarks. Starting from the severity of diabetic complications, we introduced the relationship between obesity and diabetes, transitioned to indices that can assess obesity, compared the advantages and disadvantages between WC, BMI, and WWI, and finally led to the purpose of our study.

- The authors make rather outdated statements like "BMI is the new norm for defining .....''

Response: Thank the reviewer for pointing out our oversight. In lines 68-70, we update the definition of BMI：BMI redefines “healthy weight” and “unhealthy weight” and is one of the most commonly used diagnostic tools. At the same time, more appropriate references were replaced.

- line 75 and 76 "WC is an essential part of the metabolic syndrome which also involves the detection of insulin resistance''. This sentence is confusing and should be clarified.

Response: Thanks to the reviewer for carefully pointing out our shortcomings. Because central obesity is closely related to metabolic diseases such as diabetes mellitus, hypertension, and cardiovascular and cerebrovascular diseases, and is an important contributing factor to the occurrence and development of metabolic syndrome, all of them have made central obesity as a necessary condition for the diagnosis of MS. WC can effectively respond to the degree of abdominal fat aggregation, and so it is used as a measure of central obesity[1]. In lines 74-76, we make this sentence more fluent and clear: WC plays an important role not only in the assessment of metabolic syndrome but also a central role in the detection of insulin resistance.

- Line 91 " associated with unhealthy body components..." it is body compartments not components

Response: Thank you for the linguistic details of our article. In line 91, replace "components" with " compartments".

- Line 100 "that the optimal BMI or the oldest elderly may be in the overweight or mildly obese range". This is also a very generalized and unsubstantiated statement.

Response: Sorry, our expression is not clear and accurate enough. In lines 97-99, we clarified the definition of the obesity paradox, where overweight and obese patients have a better prognosis than non-overweight or non-obese patients among those who suffer from cardiovascular disease[2].

- Line 112 "The correlation between WWI and diabetes has important clinical significance in promoting patients' physical and mental health, reducing social burden, and rationally allocating medical resources". It is unclear whether the authors are advocating for WWI as a predictor for diabetes or just correlated with diabetes. If it is the latter what is its significance then?

Response: Our study explored the correlation between WWI and diabetes and found that the correlation between WWI and diabetes was higher than the correlation between WC and BMI and diabetes. Our methodology was cross-sectional and could be used as a pilot study to guide future larger, prospectively randomised controlled studies.

2. Results

- Table 1: Did the authors stratify their study population tot ensure equal numbers among the four quartiles Q1- Q4 (N=1993). This was not indicated in their methods section.

Response: In lines 133-137, we go into detail in the methods section. In order to carry out the trend test, we used EmpowerStats software to quarter the study population according to the WWI interval and made the four groups nearly equal in size.

- Table 2: Model 3 N= 2120 while model 1 and 2 have n = 7973 participants. what happened to the missing participants in model 3?

Response: In order to analyze the relationship between WWI and diabetes in depth, we performed multivariate logistic regression analyses. Model 2 is adjusted for age, gender and ethnicity. To give stability to the results, we adjusted for confounders, which increased the number of covariates included in Model 3, including age, gender, race/ethnicity, educational level, HDL-C, LDL-C, triglyceride, total cholesterol, Family PIR, total cholesterol, marital status, hypertension status, moderate activity, protein, total sugars, total fat, smoking status and drinking alcohol status. In Model 3, the number of people decreased because more confounding factors were included, resulting in missing variables. In order to ensure accuracy, the corresponding participants were reduced.

- Figure 2 - 5. There figure legends are just the covariates assessed and not a description of the figures. The figures also lack any key and can not be interpreted.

Response: We have made changes to the legends. To test for nonlinear relationships, we performed smoothed curve fitting for WWI and diabetes. Fig 2 is the fit of the linear regression between WWI and diabetes under the fully adjusted model. Horizontal coordinates represent WWI, vertical coordinates represent diabetes. The solid red line indicates the smoothed curve fit between the variables, and the blue bands indicate the 95% confidence intervals of the fit. Fig 3 is the result of the smoothed curve fit with diabetes under the fully adjusted model after transforming WWI from a continuous variable to a quartile categorical variable. And deleted redundant fig 4 and fig 5.

3. Discussion

- The authors do not really discuss their results in relation to other published literature, the discussion section needs to be re-written. The discussion is also unfocused and delves into topics that are outside the topic of the manuscript (Lines 312 - 353)

Response: Thank you very much for your comments on the discussion section. In lines 271-253, We have rewritten the discussion section in detail. The relationship between the results and other published literature is insightfully discussed based on your comments, and have highlighted the main points of the discussion[3–9].

These comments are all valuable and enable us to greatly improve the quality of our manuscript. Our whole team made meticulous revisions to the manuscript word for word according to the reviewer's opinions, making the manuscript more fit with the title, the content more scientific and orderly, and the text expression clearer and smoother. We tried our best to improve the manuscript. These changes will not influence the content and framework of the paper. We appreciate for Editors/Reviewers’ warm work earnestly and hope that the corrections will meet with approval. Please do not hesitate to contact us with any further questions or recommendations. 

Once again, thank you very much for your comments and suggestions.

Sincerely yours, 

Jinli Guo

References

 1. Tian J, Qiu M, Li Y, Zhang X, Wang H, Sun S, et al. Contribution of birth weight and adult waist circumference to cardiovascular disease risk in a longitudinal study. Sci Rep. 2017;7: 9768. doi:10.1038/s41598-017-10176-6

2. Lavie CJ, Milani RV, Ventura HO. Obesity and cardiovascular disease: risk factor, paradox, and impact of weight loss. J Am Coll Cardiol. 2009;53: 1925–1932. doi:10.1016/j.jacc.2008.12.068

3. Xu G, Liu B, Sun Y, Du Y, Snetselaar LG, Hu FB, et al. Prevalence of diagnosed type 1 and type 2 diabetes among US adults in 2016 and 2017: population based study. BMJ. 2018;362: k1497. doi:10.1136/bmj.k1497

4. Haire-Joshu D, Glasgow RE, Tibbs TL. Smoking and diabetes. Diabetes Care. 1999;22: 1887–1898. doi:10.2337/diacare.22.11.1887

5. Yan Y, Wu T, Zhang M, Li C, Liu Q, Li F. Prevalence, awareness and control of type 2 diabetes mellitus and risk factors in Chinese elderly population. BMC Public Health. 2022;22: 1382. doi:10.1186/s12889-022-13759-9

6. Resnick HE, Valsania P, Halter JB, Lin X. Differential effects of BMI on diabetes risk among black and white Americans. Diabetes Care. 1998;21: 1828–1835. doi:10.2337/diacare.21.11.1828

7. Nakagami T, Qiao Q, Carstensen B, Nhr-Hansen C, Hu G, Tuomilehto J, et al. Age, body mass index and Type 2 diabetes-associations modified by ethnicity. Diabetologia. 2003;46: 1063–1070. doi:10.1007/s00125-003-1158-9

8. Abate N, Chandalia M. The impact of ethnicity on type 2 diabetes. J Diabetes Complications. 2003;17: 39–58. doi:10.1016/s1056-8727(02)00190-3

9. Haffner SM, Mitchell BD, Hazuda HP, Stern MP. Greater influence of central distribution of adipose tissue on incidence of non-insulin-dependent diabetes in women than men. Am J Clin Nutr. 1991;53: 1312–1317. doi:10.1093/ajcn/53.5.1312

---

## [Decision Letter · Decision Letter 2]

9 Jul 2024

PONE-D-24-02729R2Is weight-adjusted waist index more strongly associated with diabetes than body mass index and waist circumference?: results from the database large community sample studyPLOS ONE

Dear Dr. Wu,

Thank you for submitting your manuscript to PLOS ONE. After careful consideration, we feel that it has merit but does not fully meet PLOS ONE’s publication criteria as it currently stands. Therefore, we invite you to submit a revised version of the manuscript that addresses the points raised during the review process.

Please address the comment raised by the reviewerThis manuscript requires a minor revisionPlease adhere to both the journal’s general and formatting requirements ==============================

We look forward to receiving your revised manuscript.

Kind regards,

Fredirick Lazaro mashili, MD, PhD

Academic Editor

PLOS ONE

Journal Requirements:

Additional Editor Comments:

The manuscript has significantly improved, however a minor but important revision is required to qualify the manuscript for publication. Please address the comment accordingly and ensure all the formatting and general requirements are done accordingly.

Reviewers' comments:

Reviewer's Responses to Questions

**Comments to the Author**

1. If the authors have adequately addressed your comments raised in a previous round of review and you feel that this manuscript is now acceptable for publication, you may indicate that here to bypass the “Comments to the Author” section, enter your conflict of interest statement in the “Confidential to Editor” section, and submit your "Accept" recommendation.

Reviewer #3: (No Response)

2. Is the manuscript technically sound, and do the data support the conclusions?

Reviewer #3: Partly

3. Has the statistical analysis been performed appropriately and rigorously? 

Reviewer #3: No

4. Have the authors made all data underlying the findings in their manuscript fully available?

Reviewer #3: No

5. Is the manuscript presented in an intelligible fashion and written in standard English?

Reviewer #3: No

6. Review Comments to the Author

Reviewer #3: Results:

In Table 1 the authors report to have compared the categorial variables using chi squared test. This is not correct as this test is used to compare only two categorical variables. ANOVA would be a better statistical method.

In the linear regression analysis, the authors have now removed the number of participants (N = ), why is this the case?

Discussion:

The discussion is still unfocused and the authors do not really discuss the implications of their results in relation to current practice.

7. PLOS authors have the option to publish the peer review history of their article (what does this mean?). If published, this will include your full peer review and any attached files.

Reviewer #3: No

---

## [Author Response · Author response to Decision Letter 2]

15 Jul 2024

Dear Editor and Reviewers, 

We appreciate the opportunity to allow us to revise our manuscript and thanks for the reviewers’ constructive comments and suggestions. We would like to submit our revised manuscript, entitled “Is weight-adjusted waist index more strongly associated with diabetes than body mass index and waist circumference?: results from the database large community sample study” for consideration for publication. In the revised manuscript, we have carefully addressed all comments and questions raised by reviewers point-by-point. We marked the changes in red in the revised paper. We greatly appreciate your time and efforts to improve our manuscript for publication.

Reviewer #3: Results:

In Table 1 the authors report to have compared the categorial variables using chi squared test. This is not correct as this test is used to compare only two categorical variables. ANOVA would be a better statistical method.

Response: We thank the reviewer for the valuable suggestion on the article at the statistical level, you are correct in your suggestion. In our study, the weighted chi-square test was used for the comparison between categorical variables, and for the comparison between continuous variables, ANOVA was used. In fact, our study did follow the statistical methodology described above. We apologize for this misunderstanding due to an oversight in our presentation. In Table 1, we restate this part.

-In the linear regression analysis, the authors have now removed the number of participants (N = ), why is this the case?

Response: Thank you for your meticulous review. After a unanimous consultation with our team, and taking into account the published literature[1–3], it was felt that Tables 2 and 3 contained too much redundant information, so the number of participants was deleted and a uniform statement was made in the Statistical Methods section.

- The discussion is still unfocused and the authors do not really discuss the implications of their results in relation to current practice.

Response: Thank you very much for your comments on the Discussion section. Our team reviewed the content of the Discussion section and removed some non-diabetes related statements (e.g., hypertension, hyperlipidemia, cardiovascular disease, and cerebrovascular disease in lines 305-307), as well as removing content not related to preventing the onset of diabetes in lines 315-317, allowing for more focused content. A previous study of diabetes prediction methods was added to lines 323-333 of the manuscript and compared to our study, discussing the implications of our findings for current practice.

These comments are all valuable and enable us to greatly improve the quality of our manuscript. Our whole team made meticulous revisions to the manuscript word for word according to the reviewer's opinions, making the manuscript more fit with the title, the content more scientific and orderly, and the text expression clearer and smoother. We tried our best to improve the manuscript. These changes will not influence the content and framework of the paper. We appreciate for Editors/Reviewers’ warm work earnestly and hope that the corrections will meet with approval. Please do not hesitate to contact us with any further questions or recommendations. 

Once again, thank you very much for your comments and suggestions.

Sincerely yours, 

Jinli Guo

References

1. Liu Z, Kuo P-L, Horvath S, Crimmins E, Ferrucci L, Levine M. A new aging measure captures morbidity and mortality risk across diverse subpopulations from NHANES IV: A cohort study. PLoS Med. 2018;15: e1002718. doi:10.1371/journal.pmed.1002718

2. Zhang Y, Wu H, Li C, Liu C, Liu M, Liu X, et al. Associations between weight-adjusted waist index and bone mineral density: results of a nationwide survey. BMC Endocr Disord. 2023;23: 162. doi:10.1186/s12902-023-01418-y

3. Qin Y, Qiao Y, Wang D, Li M, Yang Z, Li L, et al. Visceral adiposity index is positively associated with fasting plasma glucose: a cross-sectional study from National Health and Nutrition Examination Survey 2017-2020. BMC Public Health. 2023;23: 313. doi:10.1186/s12889-023-15231-8

---

## [Decision Letter · Decision Letter 3]

7 Aug 2024

Is weight-adjusted waist index more strongly associated with diabetes than body mass index and waist circumference?: results from the database large community sample study

PONE-D-24-02729R3

Dear Dr. Wu,

We’re pleased to inform you that your manuscript has been judged scientifically suitable for publication and will be formally accepted for publication once it meets all outstanding technical requirements.

Kind regards,

Fredirick Lazaro mashili, MD, PhD

Academic Editor

PLOS ONE

Additional Editor Comments (optional):

All the comments and concerns have been taken onboard and addressed respectively. The manuscript now looks good for publication

Reviewers' comments:

Reviewer's Responses to Questions

**Comments to the Author**

1. If the authors have adequately addressed your comments raised in a previous round of review and you feel that this manuscript is now acceptable for publication, you may indicate that here to bypass the “Comments to the Author” section, enter your conflict of interest statement in the “Confidential to Editor” section, and submit your "Accept" recommendation.

Reviewer #1: All comments have been addressed

Reviewer #3: All comments have been addressed

2. Is the manuscript technically sound, and do the data support the conclusions?

Reviewer #1: Yes

Reviewer #3: Yes

3. Has the statistical analysis been performed appropriately and rigorously? 

Reviewer #1: Yes

Reviewer #3: I Don't Know

4. Have the authors made all data underlying the findings in their manuscript fully available?

Reviewer #1: Yes

Reviewer #3: Yes

5. Is the manuscript presented in an intelligible fashion and written in standard English?

Reviewer #1: Yes

Reviewer #3: Yes

6. Review Comments to the Author

Reviewer #1: The authors have now addressed all the comments raised by the reviewers. All the questions have been answered.

Reviewer #3: (No Response)

7. PLOS authors have the option to publish the peer review history of their article (what does this mean?). If published, this will include your full peer review and any attached files.

Reviewer #1: **Yes: **Fredirick mashili

Reviewer #3: No

---

## [Editor Report · Acceptance letter]

9 Aug 2024

PONE-D-24-02729R3 

PLOS ONE

Dear Dr. Wu, 

I'm pleased to inform you that your manuscript has been deemed suitable for publication in PLOS ONE. Congratulations! Your manuscript is now being handed over to our production team.

Kind regards, 

on behalf of

Dr Fredirick Lazaro mashili 

Academic Editor

PLOS ONE